# FILTER TRAINING AND MAXIMUM RESPONSE: CLASSIFICATION VIA DISCERNING

## ABSTRACT

This report introduces a training and recognition scheme, in which classification is realized via class-wise discerning. Trained with datasets whose labels are randomly shuffled except for one class of interest, a neural network learns class-wise parameter values, and remolds itself from a feature sorter into feature filters, each of which discerns objects belonging to one of the classes only. Classification of an input can be inferred from the maximum response of the filters. A multiple check with multiple versions of filters can diminish fluctuation and yields better performance. This scheme of discerning, maximum response and multiple check is a method of general viability to improve performance of feedforward networks, and the filter training itself is a promising feature abstraction procedure. In contrast to the direct sorting, the scheme mimics the classification process mediated by a series of one component picking.

## 1 INTRODUCTION

Suppose given a task of sorting a bunch of colored balls, we can usually do it in two ways. One is to randomly pick up a ball and deposit it to one of the groups according to its color. Or we can collect balls of the same color at a time until the last two colors are separated. Standard neural networks for classification tasks work in the former manner, for which it is an implied principle that every class should be on an equal footing and biases are harmful. This is the reason why the training data contain subsets of identical sample number or close in size for each class. The scheme introduced in this work is an analogue of the latter, in which the networks do not respond equally to features of different classes. Rather, they function more like filters, and ascription of an object is inferred from how strong their responses are.

The filter networks are obtained by filter training, from which we have a group of networks or more specifically a batch of parameter values, number of which is equal to the number of classes. Ensemblization appears to be a built-in trait of this recognition scheme. Since a prediction should not refer to a preassigned label, a quantitative evaluation of how each filter responds to an input should be defined on an equal footing. As long as one of the alternative filters scores higher than the correct one, the prediction is wrong. Fluctuation makes the correct filter the weaker in this one to many contest. To do it a favour we introduce another hierarchy of ensemblization that is a batch of versions for the filters, in the hope that the correct filter could be the overall winner in this tournament. These three steps constitute a classification procedure of discerning, maximum response, and multiple check (DMM).

The DMM scheme can improve accuracy of mediocre networks, networks already having high accuracy, and those trained with small-scale data. Fundamental reason for the increase is that in the filter training a multiclass problem is reduced to a pseudo binary classification, the class of interest and the others. Intuitively, we have the feeling that telling a component from a mixture is easier than sorting all the ingredients. The pseudo reduction mitigates workload. Depending on capacity of networks and amount of training samples, the increase of accuracy varies. Nevertheless, as performance improvement due to the mitigation is almost sure, the scheme can be a general route to enhance feedforward networks. Moreover, since a filter is specifically trained for one class, the filter training is a feature abstraction procedure in itself.

In the following, we first investigate the mechanism of filter training and how it works through a toy model. From this classification of points in a 2D plane, we can clearly view how the decision

boundaries are reshaped. Then, we give a probabilistic argument why the maximum response is a proper criterion to infer classification. After remarking on its relation with related works, we experiment MMD on the CIFAR-10 and MNIST datasets.

## 2 FILTER TRAINING

The training dataset of our toy model consists of randomly scattered points around four centers at $(0,0), (0,3), (3,0), (3,3)$, with a quota of $100, 70, 30, 100$ points, respectively. All directions are equally probable, and the radiuses obey a normal distribution with standard deviation of $2.0$. Points around $(0,0)$ is labelled red, green for $(0,3), (3,0)$, and blue for $(3,3)$, so that each class has 100 points. This simple classification is handled by a network made of an input and an output layer, which are fully connected layers. Size of the input layer is set as $128$ nodes, and the ReLU activation is used. The Adam optimizer with a learning rate of $10^{-4}$ and random shuffling of samples are adopted for all the trainings in this work. The ordinary training is run for $10^4$ iterations with batch size of $100$ to build the original classification network.

By filter training we want to make features pertaining to a specified class stand out and those related to the others suppressed. The most direct attempt is to train/retrain the network with the sub dataset of the class. The resulting model, however, is a nearly constant mapping to that class label, no matter what the real ascription is. In Fig. 1 are results of retraining the network with the blue subset. Fig. 1(a) is the portion of samples assigned to their original class (training accuracy), and Fig. 1(b) shows portion of the red and green data points that are assigned to blue (misassignment). In Fig. 1(c) the light blue background fully covers the dataset, indicating that no decision boundary remains. Indeed, this retrain procedure leads to a constant mapping, which blurs all the features, and the model totally loses recognition capacity.

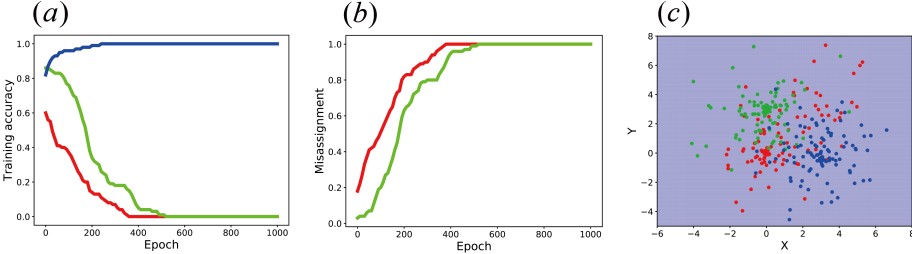

Figure 1: (a) Training accuracy for the targeted class (blue) approach 1.0, but (b) the red and green are also assigned to blue, mistakenly. (c) No decision boundary exists.

The above results are conceptually reasonable, since filtering makes no sense without unwanted components as reference. The lesson is that for filter training the targeted class along is not enough and the alternative classes should be taken into account. Accordingly, we found two viable ways to prepare the training dataset: 1) perform a random label shuffling among samples in the unspecified classes; 2) each sample, except those in the specified class, is assigned labels of all the other unspecified classes beside its own. If the random sample shuffling is used during training and the label shuffling in preparation-1 is redone at each epoch, the two are equivalent.

Datasets for training a filter of the blue are illustrated in Fig. 2. An obvious advantage of preparation-1 is smaller data size, which is equal to that of the original. For our toy model, there are only two alternative classes, so the data size is only doubled in Fig. 2(c). Size enlargement with preparation-2 is much severe when the number of classes increases. For a training set having $N_c$ classes, under each label of the alternative classes all the samples except those of the specified class should be included, so every subclass dataset is augmented by a factor of $N_c - 1$. So is the total training dataset, if $N_c - 2$ copies of the specified class is included to keep its sample density, as what is done for the toy model. For this reason, we use the equivalent reshuffling to implement preparation-2 in our experiments. Preparation-2 has its advantage that the data are better randomized by multiple labelling while preparation-1 is a dataset with definite labelling.

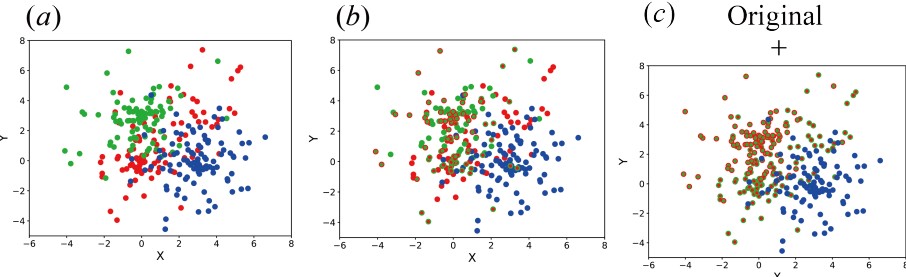

Figure 2: (a) The original training dataset. To train a filter for the blue, we can (b) randomly exchange the labels among the red and green points, as denoted by the edge color, or (c) add greenly labeled red, redly labeled green, and double the blue to keep the sample density.

Fig. 3 shows results of two independent filter trainings using preparation-1 and -2, respectively. Both the preparation procedures lead to increased accuracy of the targeted class, while accuracy for the alternative classes are decreased. More importantly, the filter training does not suffer from decision boundary destruction. A point of possible use is the similarity in the two trainings using preparation-2, where both the training accuracies and trained decision boundaries follow a similar pattern. This implies that we can use the filter training to abstract features and construct innate metrics.

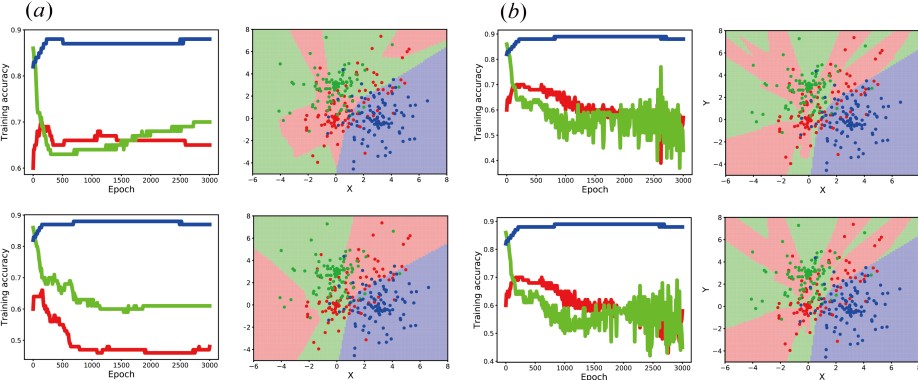

Figure 3: Training accuracy and decision boundary in two indenpendent filter trainings for the blue with (a) preparation-1 and (b) preparation-2. Stable behavor in (b) implies that filter training can reflect innate pattern of the dataset.

To see more clearly the effect of filter training, we trained a filter for the red using preparation-2 and plot in Fig. 4 snapshots of decision boundary reshaping. From comparison with Fig. 1, although the accuracy for the targeted class is less close to $100\%$, considerable increase results. In contrast to Fig. 1(b), filter training only add a little misassignment, and this small increase is understandable, since boundary expansion as well as other adaptations is a reasonable strategy for the specified class to increase its accuracy. Difference between the increase of accuracy ($\approx 0.2$) and that of misassignment ($\approx 0.1$) indicates that the drawings-in of data are purposeful and selective.

From the dynamics of decision boundary construction, we can understand how and why the filter training works. As is known, in each training step the boundary is adjusted a little bit according to the gradients to lower the cost. The gradients can be imagined as instructions encoding direction and strength of forces acting on the decision boundary. When a training batch is fed, each involved sample exerts a pull or push force on the boundary, based on its own location in the configuration space, its label, and the boundary position. Since the cost function is a mean value, forces from the samples in the unspecified classes are diminished by the random labelling, and the samples in the specified class is dominant. Still owing to the randomness, small movements by the unspecified samples in different training steps result in mutual cancelling other than increment. So only the class

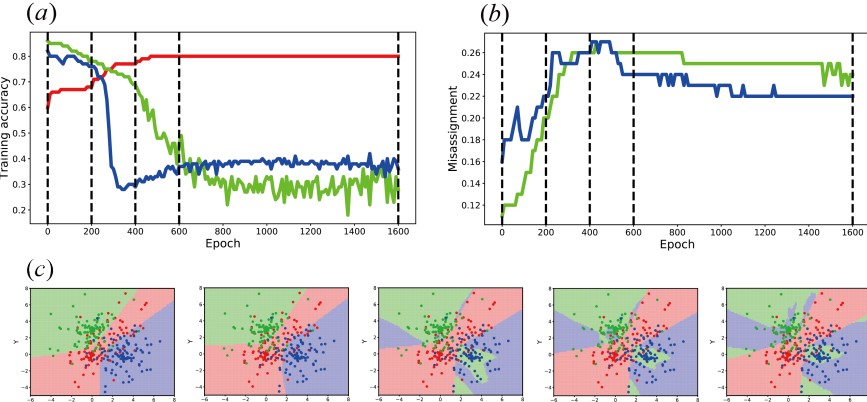

Figure 4: Snapshots of decision boundaries during fitler training for the red. Misassigment in (b) is kept relatively low. Boundary for the red is purposefully expanded to increase accuracy.

of interest accounts for effective movement of the decision boundary and adapts the boundary to a shape that well divides itself from the others.

## 3 MAXIMUM RESPONSE

Because every filter gives a softmax evaluation, we must define a selection rule to infer classification. In the ideal case, we expect only one filter has a strong response at the corresponding element, while responses of the other elements and of the other filters are suppressed. For an input belonging to the first class, for instance, it is the ideal output for the first filter that its first digit is close to $1.0$ and consequently the other digits are minimal, and meanwhile for the other filters the softmax slightly fluctuates around $1/N_c$. In reality, however, this is hardly the case.

When more than one filters give positive indication, i.e., the corresponding digit is the biggest, we choose the one having the strongest response. This criterion seems good, since it always yields a result. Nevertheless, our choice is faced with a dilemma of superiority, when outputs have a pattern as shown in Fig. 5(a), where $r_{12} > r_{11} > r_{22} > r_{21}(r_{23})$. The first filter has the strongest response at the corresponding digit ($r_{11} > r_{22}$) but the indication is negative ($r_{11} < r_{12}$), while the second filter makes a positive indication ($r_{22} > r_{21}(r_{23})$). Which is superior, the maximum response (MR) or the positive indication (PI)?

Since switch statements can not be generalized to a batch of inputs, instead of using them, we compute a prediction under the positive indication superiority by defining a score vector

$$s = p + p * m + m, \qquad (1)$$

$$p = \mathrm{agrmax}(\boldsymbol{R}) == \mathrm{agrmax}(\boldsymbol{I}), \qquad (2)$$

$$m = \frac{1}{N_c} \sum_j (\boldsymbol{R} * \boldsymbol{I})_{ij}, \qquad (3)$$

where the multiplications are elementwise. Here, $\boldsymbol{R}$ is the matrix whose rows are the softmax evaluations, and $\boldsymbol{I}$ is an $N_c \times N_c$ unit matrix whose nonzeros mark the corresponding digits. The argmax operates on the rows, so $\boldsymbol{p}$ is a vector recording the indications with 1 and 0, and $\boldsymbol{m}$ records softmax values at the corresponding digits. The second term $\boldsymbol{p} * \boldsymbol{m}$ facilitates the maximum response criterion in the case of multiple positive indications. The multiplication nulls the value of filters that give negative indications and only positive indications matter, so the superiority is untouched. The third term is added to ensure a result when no positive indication is made ($\boldsymbol{p} = 0$). Since any positive indication scores 1 in $\boldsymbol{p}$ and $\boldsymbol{m} < 1.0$, it does not alter the superiority. If only the third term presents, the maximum element of $\boldsymbol{s}$ dictates the classification, which is nothing but the maximum response criterion. It can be readily checked by substitution that only the pattern in Fig, 5(a) can trigger the difference between the two criteria.

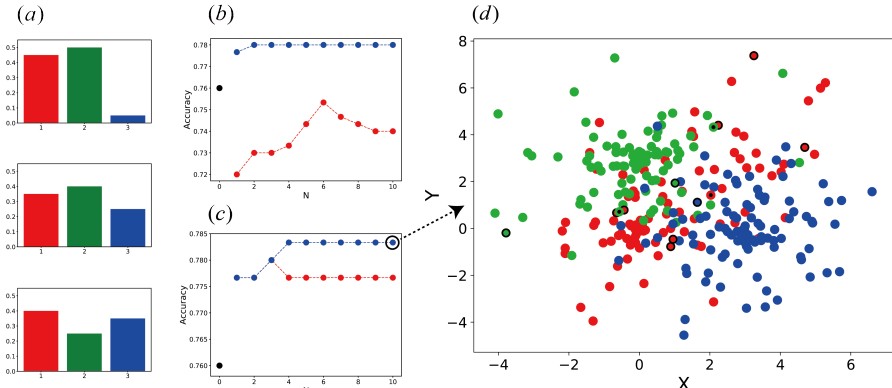

Figure 5: (a) The output pattern that causes the superiority dilimma. Multiple check results with MR (blue) and PI (red) up to 10 fitler versions for (b) preparation-1 and (c) preparation-2, where the black point denotes accuracy of the orginal network. (d) An instance of rights (circled) and wrongs (dotted) with respect to the original results: the improvements can be undertood from the view of economic brain power deployment.

Beside the maximum response, there can be other ways to solve the problem of multiple positive indications. When we are indecisive, we may consult an expert. Similarly, we can train such expert networks with only subclass datasets that are concerned, and refer to the corresponding one according to the positive indications. Another resort is the peer review like checking. Similarity in background and diversity in specialty may compensate the weakness and bias. Since this multiple check procedure is more pertinent to filtering in the sense that it can suppress fluctuations, we focus our study on it. Here two types of fluctuations should be distinguished. One is fluctuation in the softmax evaluations, which is harmful in that it can cause false positive indications. When they occurs, however, it is desirable that distribution of the false positive indications fluctuates in different filter versions. With the latter, incremental status of the real positive indication could be built up. This is the mechanism of the multiple check, and the motivation of introducing randomness.

As randomness has already been there in the training process and data preparation, in the multiple check we just retrain the network to build new versions, and scores of different versions are averaged. For the toy model, we use the training data also as test data and plot in Fig. 5 the test accuracy up to 10 filter versions. Better performance for preparation-2 (Fig. 5(c)) suggests that fluctuation is indeed harmful. Stronger fluctuations are also the cause for the bigger difference in Fig. 5(b), since we can see from Fig. 5(a) that occurrence of the pattern requires strong fluctuations. In both results, the maximum response is superior to the positive indication. There could be a reason.

Let us consider a point located near the decision boundary, where improvements are actually made by DMM (see Fig. 5(d)). As the softmax is unitary, we assume the Dirichlet distribution for output of every filter, when randomness are introduced into filter versions by the stochastic operations such as the stochastic gradient descent, the random shuffling of training samples, and the random label assignment in filter training. Note that the fixated point as input is unchanged while the outputs in different versions are stochastic due to the randomness. For a point exactly at the ideal boundary separating class-1 and class-2,

$$D_1^0 = \text{Dir}(\alpha, \beta, \beta, \cdots, \beta), \tag{4}$$

$$D_2^0 = \text{Dir}(\beta, \alpha, \beta, \cdots, \beta), \tag{5}$$

$$\cdots \cdots \cdots \cdots \cdots \cdots, \tag{6}$$

$$D_{N_c}^0 = \cdots \cdots \cdots \cdots \cdots, \tag{7}$$

is a proper assumption by symmetry argument. The less relevant distributions are denoted in ellipsis, for which the probability at the corresponding digit is a little bit biased, since filter training tends to expand the decision boundary enclosing the class of interest. Compared to other $r_{1j}(j \neq 1)$, $r_{12}$ is not endowed with specialty, because in the filter training for class-1, these classes ($j \neq 1$) are treated equally. It is the same for $r_{21}$.

When the point shifts away form class-2 and into class-1, a bias is raised between the two filters. The probability density peak of filter-1 should move toward the high end of $r_{11}$, and vice versa for filter-2, so we can expect

$$D_1 = \text{Dir}(\alpha + \Delta, \beta, \beta, \cdots, \beta), D_2 = \text{Dir}(\beta, \alpha - \Delta, \beta, \cdots, \beta) \tag{8}$$

with $\Delta > 0$, and for shift into class-2 the sign is reversed, $\Delta \to -\Delta$. Here, we do not alter $\beta$ and parameters of the remaining distributions (so $D_n = D_n^0, n > 2$), which implies that we attribute variation of distribution on these digits to secondary effect of varied $\alpha$. This is plausible, since $\Delta$ is a function of the direction and distance the point away from the boundary, and meanwhile labels for the alternative classes are randomly shuffled, which diminishes dependence on position and direction. Of course, weak dependence could remain, since the training data themselves are not evenly distributed in the configuration space.

For clarity and without loss of generality, we can assume $r_{ik} > r_{ij}$ for $k > j$, beside $r_{12} > r_{11} > r_{22} > r_{21}$. Then, incidence of the pattern is given by integral $P = \int D_1 D_2 \cdots D_{N_c} d\boldsymbol{R}$ over the assumed region subject to unity $\sum_j r_{ij} = 1$. Now, the problem is a Bayesian like inference—shift into which class yields higher incidence. As what really matters is the trend, we can avoid tedious calculation for an analytic formulation by taking the derivative with respect to $\Delta$. At the boundary $\Delta = 0$, and presence of both $\Delta$ and $-\Delta$ cancels the derivatives of the normalization factors. The resulting derivative reads

$$\left.\frac{dP}{d\Delta}\right|_{\Delta=0} = \int \ln\left(\frac{r_{11}}{r_{22}}\right) D_1^0 D_2^0 \cdots D_{N_c}^0 d\boldsymbol{R}, \tag{9}$$

which is positive due to $r_{11} > r_{22}$. So a positive $\Delta$, corresponding to shift into class-1, increases the incidence. This means that an input having the output in Fig. 5(a) should be ascribed to class-1 and hence maximum response is superior to positive indication.

Conceptually, the enhancement by DMM is due to the mitigation of workload, so that more delicate decision boundaries can be constructed. With the wrongs and rights with respect to the original results as shown in Fig. 5(d), we can go to more details. Near where DMM makes a mistake, a correction is made to compensate, which ensures an accuracy not going lower. The three close points around $(-0.5, -1.5)$ is an instance that one wrong is traded for two rights and has one net gain. The closeness of the points implies that high expense of capacity is required to make a delicate discrimination. It is like that a person needs to spend considerable brain power to obtain relatively small gain, which is uneconomic if the brain power is not abundant. This economic consideration also shows up in the sparse regions, where DMM adds rights but no wrongs. Due to the sparsity, a gain requires paying more attention to adapt the decision boundary more, which is also an uneconomic deployment without the release of brain power.

## 4 RELATED WORKS

Basically, DMM is an ensemble learning scheme (Dasarathy & Sheela, 1979; Hansen & Salamon, 1990; Wolpert, 1992) with specialties in several aspects. An obvious distinction is that one filter can not do classification but a class specified evaluation of pertinence. While each sub network in the main stream ensemble learning schemes is a classifier, only a full batch of filters constitutes one. The maximum response use the classifier selection approach (Jacobs et al., 1991; Jordan & Jacobs, 1994; Woods et al., 1997; Alpaydin & Jordan, 1996; Giacinto & Roli, 2001), and the multiple check is a classifier fusion approach (Cho & Kim, 1995; Kuncheva et al., 2001), so DMM integrates the two ensemble learning fashions. As the random label shuffling can be performed on any labelled dataset, one can incorporate DMM into other schemes as an additional ensemblization hierarchy.

Owing to difference between the random label shuffling and the usually used random (Breiman, 1996) and/or refining (Schapire, 1990; Freund & Schapire, 1997; Ho, 1998) resampling, in filter training the task simplification is not due to partition of the data but realized by making most of them irrelevant. Interplay between the label randomization and these resampling strategies may bring mutual benefits. Resampling can be helpful in that they are ways to provide diversity and realize feature splitting, which could yields more specific filters and diverse versions. Reciprocally, the filter training can be a tool to guide the resampling, since a filter is specified for one class but still give full softmax output, which is a measure of pertinence. In other words, by filter training we

can construct metrics of the dataset that is more specific for each class and more comprehensive in the sense that we have measures from perspectives of every class.

## 5 EXPERIMENTS

All the experiments are implemented with a CNN made of two convolutional layers followed by two fully connected layers. The convolutional layers have respectively 32 and 64 filters of size $5 \times 5$ and stride 2. The hidden fully connected layer has 128 nodes. The ReLU activation is applied to every layer (except the output one). The original network is trained for 100 epochs with batch size of 100. To exclude the possibility that the improvement is due to unfulfilled training, we check this by refining the output layer with additional $10^4$ iterations, which does not lead to improvement. All the filter trainings in the experiments also run for $10^4$ iterations. The training accuracy and misassignment are evaluated with the training batch to be fed. As fine tuning the last layers of a pretrained model can usually adapt it to a new task, in filter training we only retrain the output layer. This is not necessarily a standard, since filter training can be performed on the whole network or any part of it.

For CIFAR-10, at first we directly performed filter training to the output layer, and the results are shown in Fig. 6(a). The increasing accuracy for the specified class, the falling accuracy for an alternative class, and the low misassignment all suggest that the filter training works well. But the multiple check results with 10 independent implementations of preparation-1 does not lead to an increment. When the training samples are abundant, the random label shuffling is not random enough to disperse the distribution of false positive indications, so that the different filter versions tend to make similar mistakes. Similar accuracies for the two criteria are an indication of small fluctuation in the softmax outputs, since, as we have noted, fluctuation facilitates the outputs that cause the difference.

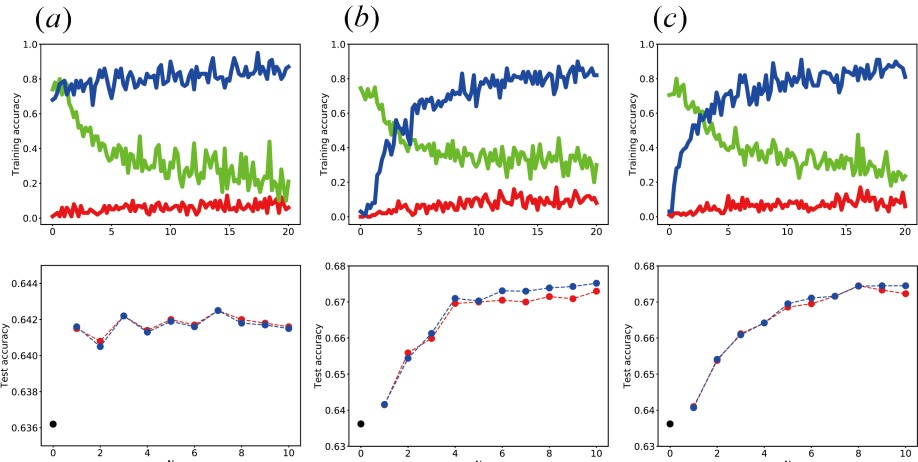

Figure 6: Upper: typical training accuracies for the specified class (blue) and for an alternative class (green), and misassignment (red) on CIFAR-10, with (a) preparation-1, (b) class-wise label permutation and preparation-1, and (c) class-wise label permutation and preparation-2. Down: mutiple check results with MR (blue) and PI (red), and the original test accuracy is marked for comparison.

We tried to facilitate diversity in different filter versions by randomly permuting the class labels and retraining the last two layers with $10^4$ iterations, and then remolded the output layer to filters. We note that this permutation and the random shuffling in preparation-1 are utterly different in that the former is class-wise while the latter is sample-wise. From multiple check results in Fig. 6(b), the strategy successfully builds up an increment. Start of the training accuracy for the specified class is low, because the label permutation is purposefully misleading. In Fig. 6(c), we changed preparation-1 to preparation-2, which does not improve the accuracy further. The takeaway is that more randomization among filter versions is needed when the data size is large, and since the sample

abundance has largely diminished randomness within a softmax evaluation, preparation-2 adds little benefit. We used preparation-1 in the remaining experiments.

We experimented small size training data by using the first 500 samples for each class and present the results in Fig. 7(a). Here, the single version result is worse than the original. The reason could be overfitting caused by the filter training, which is corrected by the multiple check. In Fig. 7(b), we divided the full CIFAR-10 dataset into 10 partitions and trained 10 filter versions, with which the accuracy is further increased. We turned off the class-wise label permutation in Fig. 7(c), which does not worsen the performance. This implies the possibility that the diversity can result from modification of the dataset itself, such as augmentation or randomization with noise.

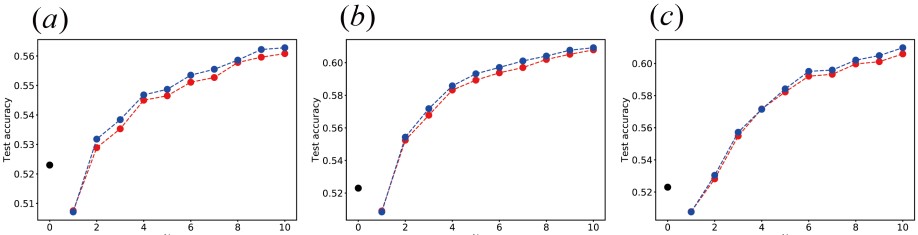

Figure 7: Mutiple check results for reduced CIFAR-10 with (a) a single partition (b) full partitions and the class-wise label permutation, and (c) full partitions without the label permutation.

From results for MNIST in Fig. 8, both the training accuracy for the specified class and the test accuracies are increased, despite small room for improvement. Moreover, the single version improvement is the biggest in all the experiments. We can expect networks that already have high accuracy to work quite well with DMM, since there is no reason for worse performance, when the capacity is relatively strong and the task becomes easier.

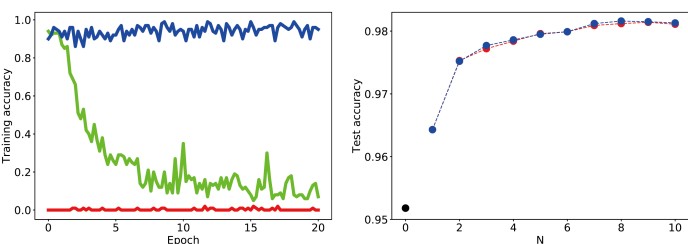

Figure 8: Results for the MNIST dataset with preparation-1 and the class-wise label permutation. Notations are the same as the above.

## 6    CONCLUSION

Motivated by the classification process via one component discerning, we propose the filter training and maximum response. The multiple check can build up an increment of performance if the fluctuation in the responses are properly distributed among filter versions. How to deal with the two types of fluctuations is the major concern for it to work well. DMM constitutes a special ensemble learning scheme, which itself can be incorporated into other schemes as an additional hierarchy of ensemblization. It is beneficial if similar mechanism can be integrated into other network architectures, since task simplification is a common strategy in intelligence activities.

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
