# OpenReview forum: "Filter Training and Maximum Response: Classification via Discerning"
_ICLR.cc/2019/Conference_

### Official Review · AnonReviewer3 · 2018-10-16
**A new DMM learning scheme is proposed to enhance classification network. The work has acceptable quality and clarity. Some more discussion and comparison to recent classification network is needed.**

**Rating:** 6
**Confidence:** 3

**Review:**

The author proposed a novel classification procedure of discerning, maximum response, and multiple check (DMM) to improve accuracy of mediocre network and enhance feedforward network. The author used a toy model with randomly scattered points around four centres to demonstrate the dynamics of decision boundary construction and show how and why the filter training had worked when a multiclass problem is reduced to a pseudo binary classification. The literature review is rather short, some discussion on how the proposed work advances the latest deep neural networks should be added. The experiments are carried out with a simple CNN, it will be better if the author demonstrate its integration to more state-of-art network and make a comparison to their performance.

---

### Official Review · AnonReviewer2 · 2018-10-30
**No real distinction from classical One vs. All classification**

**Rating:** 3
**Confidence:** 4

**Review:**

The paper discusses a method to increase accuracy of deep-nets on multi-class classification tasks by what seems to be a reduction of multi-class to binary classification following the classical one-vs-all mechanism. I fail to see any novelty in the paper. The problem of reducing multi-class to binary classification has been studied thoroughly with many classical papers like:

1. Usual one-vs-all - this paper does the same thing as one vs all in my opinion even though this technique is known for a decade or so.
2. Weighted One-Against-All - http://hunch.net/~jl/projects/reductions/woa/woa.pdf

and more sophisticated techniques like:

3. Alina Beygelzimer, John Langford, Pradeep D. Ravikumar. Error-Correcting Tournaments. CoRR, abs/0902.3176, 2009.
4. Erin L. Allwein, Robert E. Schapire, Yoram Singer, Pack Kaelbling. Reducing Multiclass to Binary: A Unifying Approach for Margin Classifiers. Journal of Machine Learning Research, 113—141, 2000.
5. Thomas G. Dietterich, Ghulum Bakiri. Solving multiclass learning problems via error-correcting output codes. Journal of Artificial Intelligence Research, 2:263—286, 1995.

In my opinion the methods in [1,2,3,5] above can be used with any binary learners and therefore deep-networks. This paper makes no effort in comparing with any of these well-known papers. Moreover the experiments do not show any gain in state of the art performances in the data-sets used, as experiments are done with toy-networks. Further some rules for selecting the class is discussed in Section 3. There are many known rules for generating probability scores in one-vs-all classification and the relations to these are not discussed.

Therefore, I fail to see any novelty in this paper (theoretical or empirical).

---

### Official Review · AnonReviewer1 · 2018-11-01
**I am not able to assess this work**

**Rating:** 2
**Confidence:** 1

**Review:**

Unfortunately I don't understand what this paper is about. Please assign to another reviewer.

---

### Meta-Review · Area_Chair1 · 2018-12-17
**Submission lacking grounding in previous work**

**Confidence:** 5
**Recommendation:** Reject

**Metareview:**

This work examines how to deal with multiple classes. Unfortunately, as reviewers note, it fails to adequately ground its approach in previous work and show how the architecture relates to the considerable research that has examined the question beforehand.